# Investigation of Silicon Core-Based Fiber Bragg Grating for Simultaneous Detection of Temperature and Refractive Index

**DOI:** 10.3390/s23083936

**Published:** 2023-04-12

**Authors:** Yi-Lin Yu, Yu-Hua Hong, Yu-Hsuan Chen, Hiroki Kishikawa, Kimio Oguchi

**Affiliations:** 1Department of Electrical Engineering, Feng-Chia University, Taichung 407102, Taiwan; 2Department of Optical Science, Tokushima University, Tokushima 770-0855, Japan; 3Department of Electronic and Computer Engineering, National Taiwan University of Science and Technology, Taipei City 106216, Taiwan

**Keywords:** silicon core fiber, fiber Bragg grating, fiber sensing

## Abstract

In this article, we theoretically designed and simulated a silicon core fiber for the simultaneous detection of temperature and refractive index. We first discussed the parameters of the silicon core fiber for near single-mode operation. Second, we designed and simulated a silicon core-based fiber Bragg grating and applied it for simultaneous sensing of temperature and environmental refractive index. The sensitivities for the temperature and refractive index were 80.5 pm/°C and 208.76 dB/RIU, respectively, within a temperature range of 0 to 50 °C and a refractive index range of 1.0 to 1.4. The proposed fiber sensor head can provide a method with simple structure and high sensitivity for various sensing targets.

## 1. Introduction

Sensing technologies based on optical fibers have several inherent advantages that make them attractive for a wide range of industrial sensing applications [1,2]. They are typically small in size, passive, immune to electromagnetic interference, resistant to harsh environments, and have the ability to perform distributed sensing [3,4]. Although there are many different fiber optic sensor technologies (e.g., fiber-based interferometers [5,6]), the most frequently used fiber-based sensing device at the present time is fiber Bragg grating (FBG). FBG is a multifunctional fiber device that is applied for communication and sensing because of the mature manufacturing technology and low cost compared with other fiber devices [7]. In the past decade, semiconductor-based core fibers have become more and more attractive [8], especially those based on silicon materials [9]. Most of the recent research into silicon core-based fibers shows nonlinear effects because they have a higher nonlinear coefficient than fused silica, e.g., four-wave mixing and supercontinuum generation [10,11]. Compared to fused silica fibers, the core diameter of SCFs needed to be reduced to smaller than 1 μm for single-mode operation [12]; in other words, SCFs has more potential applications for nonlinear effect. Apart from nonlinear effects, U. J. Gibson used a 30 W CO_2_ laser to fabricate silicon core fiber Bragg grating in 2017 [13]. During the manufacturing process, they controlled the laser power by using 5 kHz pulse-width modulation and a ZnSe lens. In addition, the authors used the proposed silicon-core FBG to independently detect the temperature variation from 30 to 55 °C with a sensitivity of 76.64 pm/°C, and an axial strain from 0 to 2600 με with a sensitivity of around 0.235 pm/με [14]. In reality, the connection between SCF and fused silica-based fiber was an important issue that needed to be overcome. Wang et al. [15] proposed that the SCF could be connected to fused silica-based fiber devices via a mid-IR single-mode fiber (SMF) to a tapered SCF. In addition, there were other techniques for avoiding exciting higher-order modes when coupling a Gaussian beam to a different type of fiber, such as offset launch technique [16] and mode field-matched center launching [17].

In this manuscript, we proposed a silicon core-based FBG that could be applied for simultaneous temperature and refractive index sensing. The purpose of this was because the refractive index of a material is changed with temperature. Therefore, the simultaneous detection of temperature and refractive index is necessary when monitoring a target. If the impact of environmental refractive index is negligible, the proposed fiber sensor head could be applied for harsh temperature environments, e.g., nuclear reactors [18]. When the sensor head was applied for detecting temperature and refractive index simultaneously, the sensing ranges for temperature and refractive index were from 0 to 50 °C and 1.0 to 1.4, respectively. However, we also theoretically discussed the characteristic of SCF, including single- or multi-mode operation with a different radius of SCF compared to our original research [19]. This condition was important; multimode operation of the silicon core-based FBG would cause more than two reflection wavelengths leading to misjudgment or confusion for the user. Due to the proposed method for data analysis, the temperature sensing was limited. However, the sensing range for temperature and refractive index still covered a majority of substances, e.g., ethanol, methanol, and so on.

## 2. Materials and Methods

Figure 1a shows the 3D model of the proposed silicon core-based fiber Bragg grating; the length of Si-FBG was set as 1200 µm, which was long enough for fiber sensing [20]. In addition, the detail of Si-FBG is presented in Figure 1b, which shows the cross-section. The diameter of the silicon core and glass-based cladding were initially set at 1 μm (*D_c_*) and 12 μm (*D_cl_*), respectively, and the period of Si-FBG was 1760 nm (*Λ*). We set the glass-based cladding diameter to 12 μm to decrease the simulation period. However, there were no obviously different results between 12 μm and other larger cladding diameter sizes (e.g., 125 μm diameter), especially for reflectance. According to past research [12], the main condition for single-mode operation of SCF is that the core diameter of SCF should be smaller than 1 µm. To investigate this condition, we simulated transmission transparency with a different radius of silicon core, from 0.45 to 0.75, and transmission mode, respectively, without fiber Bragg grating design. In addition, in the material setting, the dispersion property both of silica and silicon of the proposed schematic can be described by the Sellmeier equation [21]:(1)$$n^{2}\left(\lambda \right)=1+\frac{A_{1}\lambda^{2}}{\lambda^{2}-B_{1}}+\frac{A_{2}\lambda^{2}}{\lambda^{2}-B_{2}}+\frac{A_{3}\lambda^{2}}{\lambda^{2}-B_{3}}$$
where *n* and λ are the refractive index value and operating wavelength, respectively. In the silica case, the coefficients are A_1_ = 0.696166, A_2_ = 0.4079426, A_3_ = 0.8974794, B_1_ = 4.6791482 × 10^−3^ μm^2^, B_2_ = 1.351206 × 10^−2^ μm^2^, B_3_ = 97.934 μm^2^ [21]. In the silicon case, the coefficients are A_1_ = 10.66842, A_2_ = 0.003043, A_3_ = 1.54133, B_1_ = 9.09122 × 10^−2^ μm^2^, B_2_ = 1.2876602 μm^2^, B_3_ = 1.21882 × 10^6^ μm^2^ [22]. The initial results are shown in Figure 2a–d with different r_c_ and transmission mode. According to the results, the cut-off wavelength of TE02 mode would be shifted toward long wavelength if the r_c_ size increased, and the inset in Figure 2a–d shows the transmission profiles with different transmission mode at cut-off wavelengths of 1397 nm, 1549 nm, 2006 nm, and 2310 nm, respectively. In the 0.45-r_c_ case, the transmission profile of TE02 mode was obviously different to other cases, which was because of the smaller size of r_c_. The V-number (also called normalized frequency) of a fiber is a useful specification for the number of transmission modes at a given wavelength, which could be calculated by [23]
(2)$$V=\frac{2\pi \cdot r_{c}}{\lambda }\sqrt{n_{c}^{2}-n_{cl}^{2}}=\frac{2\pi \cdot r_{c}}{\lambda }NA$$
where the *V* is the V-number, *λ* is the wavelength, n_c_ is the refractive index of the core, n_cl_ is the refractive index of the cladding, and NA is the numerical aperture of the fiber. The calculation results for V-number are shown in Figure 2e. The black solid and dashed lines show that the V-number decreases when the operating wavelength increases to a longer wavelength. The red solid line is the limitation for near single-mode operation, which means a V-number below the red solid line is necessary for near single mode, and the value was around 6.4. In Figure 2a–d, TE01 and TM01 mode were still generated simultaneously when the operating wavelength was shorter than the cut-off wavelength. However, TE01 mode had more efficient transmission than TM01 due to the lower loss, which was so-called near single-mode operation in this study. Figure 2f shows the transmission efficiency with different r_c_ and modes. To verify the reflection wavelength, we also simulated reflected output spectrum at 25 °C environmental temperature with the different r_c_ and transmission modes. The results are shown in Figure 3a–d. In the cases of 0.6 and 0.75 μm-r_c_, the TE02 mode was generated in the communication wavelength band, which could cause user misjudgment, so the larger r_c_ should be avoided. Second, the reflected wavelength of the TE01 mode in the 0.5 μm-r_c_ case was 1549.43 nm, which was close to the end region of the TE02 mode at around 1549 nm; if the regions are too close, this limits the temperature sensing. The third case is the 0.45-r_c_ case, which had a good spectrum for sensing applications. However, the smaller r_c_ could cause insensitivity for the refractive index sensing because of the smaller contact area between the silicon core and environmental medium. To solve the problem in the second case (0.5 μm-r_c_), we could easily increase the grating period to 1794 to shift the reflected central wavelength to 1569.8 nm with around 0.65 reflectivity. During our simulation process, the records show that the relationship between period and reflected central wavelength could be written as
(3)$$\lambda \left(\mathrm{nm}\right)=0.599\cdot \Lambda +494.983$$

The updated reflection spectrum is shown in Figure 3e, and the central wavelength was still within the communication band. The central wavelength λ_B_ was related to grating period, which is defined as a Bragg condition, and can be written as [24]
(4)$$\lambda_{B}=2\cdot n_{eff}\cdot \Lambda$$
where the n_eff_ is effective refractive index of the waveguide. According to ref. [13], the central wavelength corresponded to an 8^th^ order grating of the proposed Si-FBG. To investigate the resulting order number of grating, we collected the output spectrum of TE01 mode from Figure 3a–d. The results in Figure 3f show the relationship between grating order and wavelength, and can be described as
(5)$$\lambda_{B}=\frac{2\cdot n_{eff}\cdot \Lambda }{N}$$
where the *N* is the order number of the grating. However, it was shown that the relationship between order number of the grating and reflected central wavelength was not linear but square. The calculated electric field with TE01 mode in different cross-sections of the 3D model at the end of Si-FBG is shown in Figure 4a–d. In Figure 4a,c, the results show the low reflection of Si-FBG in the x–y and y–z cross-sections, whereas Figure 4b,d show the high reflection of Si-FBG in the x–y and y–z cross-sections, respectively.

The calculated electric field in TE01 mode with different cross-sections in the 3D model at the end of Si-FBG are shown in Figure 4a–d. In Figure 4a,c, the results show the low reflection of Si-FBG in the x–y and y–z cross-sections, whereas Figure 4b,d show the high reflection of Si-FBG in the x–y and y–z cross-sections, respectively.

## 3. Results and Discussion

a.
*Temperature sensing*


For the temperature sensing, we set the boundary condition of fiber ends as the same as the silicon core fiber to negate the environmental refractive index condition. Figure 5a shows the wavelength responses when the temperature increased or decreased, and the relationship between the peak wavelength and temperature is shown in Figure 5b with R-squared of the fitting curve of 0.9999. By means of data fitting, the regression equation can be expressed as:(6)$$\lambda \left(\mathrm{nm}\right)=0.0805\cdot T_{en}+1567.82$$
where *λ* is the peak wavelength and *T_en_* is the environmental temperature. The calculated sensitivity was 80.5 pm/°C, which is higher than the fused silica core-based FBG due to the high hermos-optic coefficient and thermal expansion coefficient of 1.8 × 10^−4^ K^−1^ and 2.6 × 10^−6^ K^−1^, respectively, compared with the case of fused silica where the values are 9.6 × 10^−6^ K^−1^ and 0.57 × 10^−6^ K^−1^, respectively.

b.
*Refractive index sensing*


For refractive index sensing, we keep the same conditions as part A, with temperature at 25 °C, and only changed the boundary condition of fiber ends with refractive index from 1.0 to 1.4. The simulation results of output spectrum are shown in Figure 6a. There was no obvious wavelength shift of central peak, which was at 1569.8 nm, because the environmental refractive index changes had no influence on the Bragg condition. The interface between a fiber and the environment formed a Fresnel reflection, which is like a partially reflecting mirror; the Fresnel equation can be defined as [25]
(7)$$R={\left(\frac{cos\theta_{i}-\frac{n_{t}}{n_{i}}cos\theta_{t}}{cos\theta_{i}+\frac{n_{t}}{n_{i}}cos\theta_{t}}\right)}^{2}$$
where the n_i_ and n_t_ are the refractive index of the silicon core and environmental refractive index, respectively, and *θ_i_* and *θ_t_* are the angle of incidence and transmission, which was 0 degrees. To compare the theoretical calculations with our simulation results, we collected the reflectance of central, first side-mode of left and first side-mode of right with different refractive indices; the results are shown in Figure 6b. All the R-squared of the fitting curves were higher than 0.9998. The trends proved that our results fitted well with theoretical calculations.

In this article, we proposed another analysis method. First, we transformed the results of Figure 6a through fast Fourier transform (FFT); the results are shown in Figure 6a. Second, we picked the first point of spatial frequency with different refractive index, and the relationship between first point of spatial frequency and different refractive index is shown in Figure 7b. The R-squared was higher than 0.9999, and the fitting curve could be written as
(8)$$FPSF\;\left(dB\right)=129.262\times n^{2}-518.652\times n+580.414$$
where FPSF represents the first point of spatial frequency and *n* is the environmental refractive index. The values of FPSF were up to 300 times larger than the reflectance in Figure 6b, which means this analysis method has higher resolution than the original analysis method.

c.
*Simultaneous sensing*


Before considering the simultaneous sensing of temperature and refractive index, we separately investigated the FPSF of these two parameters, as shown in Figure 8a. We fixed the temperature at 25 °C and 50 °C (red solid and dash lines) and changed the refractive index from 1.0 to 1.4. We also fixed the refractive index at 1.1 and 1.2, and changed the temperature from 0 to 50 °C. The trends shown were that the main condition influencing the FPSF was refractive index, and that FPSF was insensitive to temperature.

We collected all the simulation results of FPSF, and the distribution map for different temperatures and refractive indices is shown in Figure 8b. The points (X, Y, Z) indicated in the figure refer to temperature, refractive index, and FPSF, respectively.

For convenience, the overall process to identify the two parameters in simultaneous sensing can be outlined as follows:Step 1: Identify the FBG peak and read its wavelength from the spectrum in Figure 5a.Step 2: Calculate the temperature using the central wavelength and linear curve of the FBG in Figure 5b and Equation (6).Step 3: Transform the measurement of spectra by using fast Fourier transform.Step 4: Locate the spatial frequency in Figure 8b by using the calculation results from Step 3.

Finally, the performance comparison between the proposed fiber sensor and the previously reported literatures is indicated in Table 1. Although our performance was not the best, the proposed fiber sensor head was easy to fabricate compared to other devices.

## 4. Conclusions

We theoretically proposed a silicon core-based FBG for monitoring environmental temperature and refractive index. For single-mode operation in the communication band, the radius of SCF should be less than 0.5 μm. The sensing properties of the proposed sensor have been simulated by using a finite element method and a 3D model. The sensitivities of temperature and refractive index were 80.5 pm/°C and 208.76 dB/RIU, respectively, with linear regression in the temperature range from 0 to 50 °C and the refractive index range from 1.0 to 1.4. Without considering the refractive index factor, the range of temperature sensing could be extended from −100 to +700 °C, which means the proposed sensor had potential future applications. In additionally, the analyzed method could be imported into machine learning and the database used to develop artificial intelligence algorithms.

## Figures and Tables

**Figure 1 sensors-23-03936-f001:**
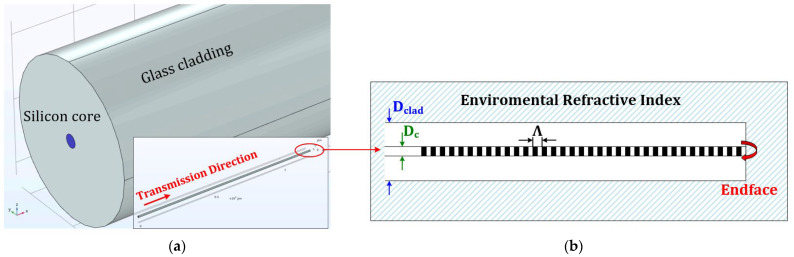
(**a**) 3D structure of fiber Bragg grating in COMSOL Multiphasic and (**b**) cross-section of silicon core-based fiber Bragg grating.

**Figure 2 sensors-23-03936-f002:**
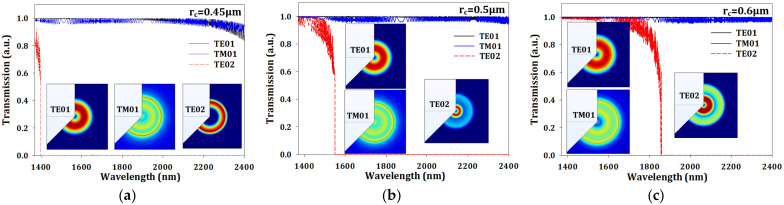
The transmission profiles with r_c_ of (**a**) 0.45 μm, (**b**) 0.5 μm, (**c**) 0.6 μm, and (**d**) 0.75 μm without grating design, (**e**) the V-number, and (**f**) the transmission in TE01 and TM01 modes with different r_c_.

**Figure 3 sensors-23-03936-f003:**
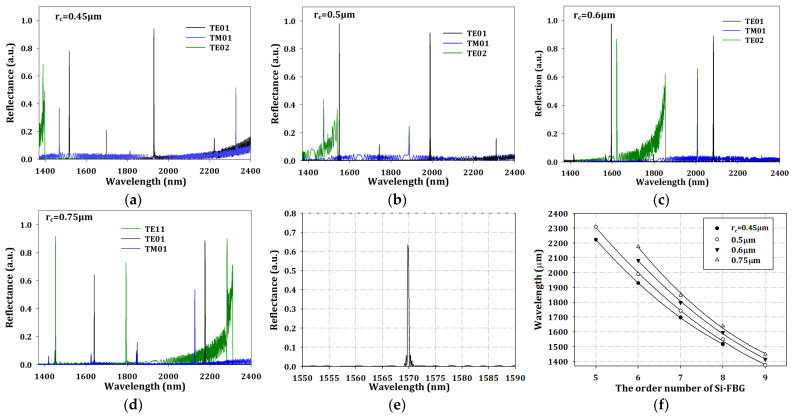
The output spectrum with different transmission modes and (**a**) r_c_-0.45 μm, (**b**) r_c_-0.5 μm, (**c**) r_c_-0.6 μm, (**d**) r_c_-0.75 μm, (**e**) the output spectrum with 1794 nm grating period, and (**f**) the trends in wavelength and order number of grating.

**Figure 4 sensors-23-03936-f004:**
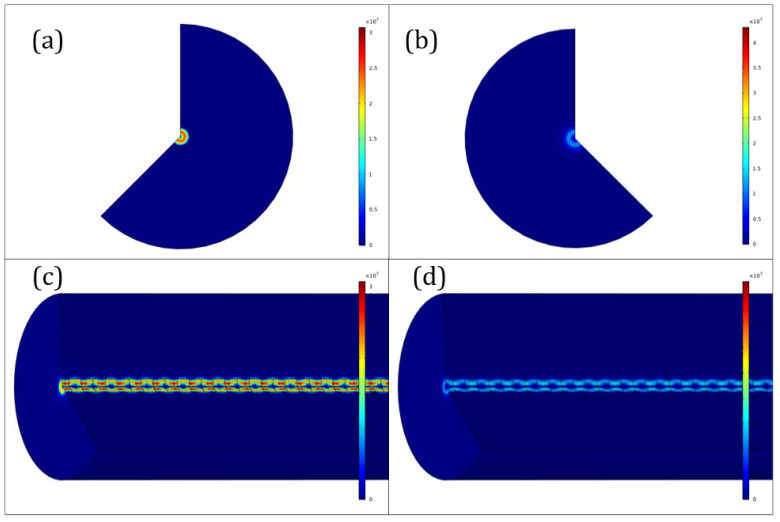
The distribution of 3D-simulated electric field in (**a**) x–y and (**c**) y–z cross-sections with low reflection and in (**b**) x–y and (**d**) y–z cross-sections with high reflection at the fiber end.

**Figure 5 sensors-23-03936-f005:**
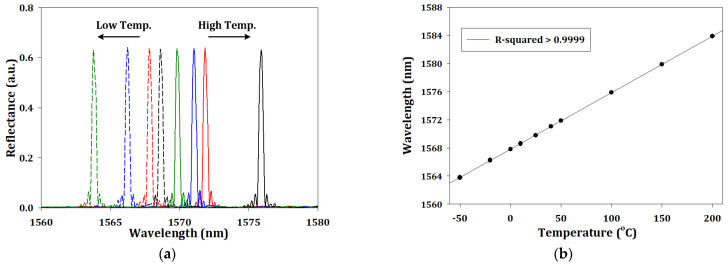
(**a**) The simulation output spectrum and (**b**) the trends in peak wavelength with different temperature.

**Figure 6 sensors-23-03936-f006:**
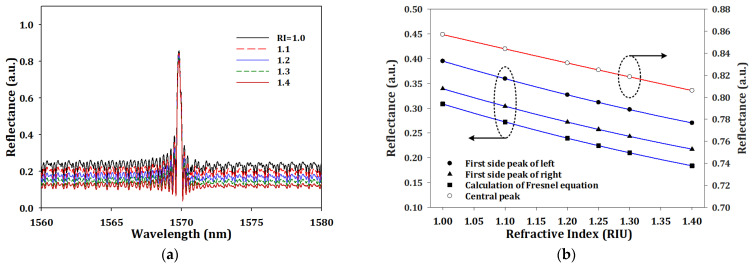
(**a**) The simulation output spectrum and (**b**) the trends in different peak wavelengths with different environmental refractive index.

**Figure 7 sensors-23-03936-f007:**
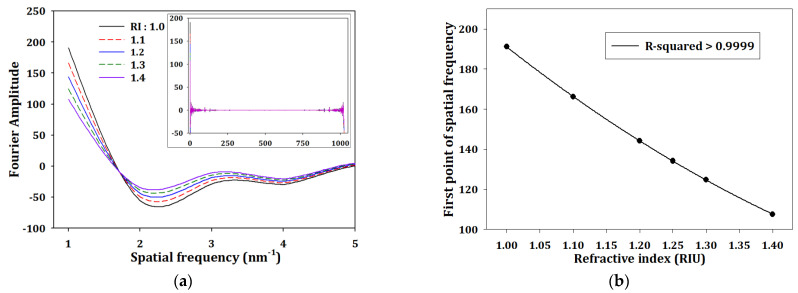
(**a**) The calculation of Fourier amplitude and (**b**) the trends in first point of spatial frequency from Figure 4a.

**Figure 8 sensors-23-03936-f008:**
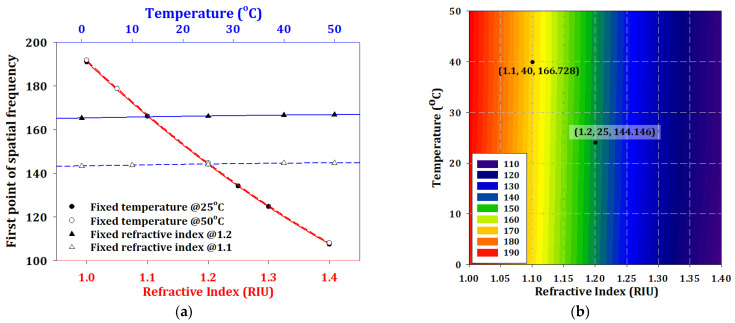
(**a**) The FPSF distribution with different temperatures and a refractive index of 1.3, (**b**) the distribution map for different temperatures and refractive indices.

**Table 1 sensors-23-03936-t001:** Comparison of simultaneous refractive index and temperature-sensing performance study with existing sensors.

Ref. (Year)	Sensor Type	Refractive Index Sensitivity	Temperature Sensitivity
[26] 2022	1. Anti-resonant reflection 2. Waveguide fiber ring-shaped structure	108.61 nm/RIU	19 pm/°C
[27] 2022	1. Michelson interferometer 2. Waist-enlarged fiber bitaper	−191.06 dBm/RIU	0.12 nm/°C
[28] 2022	1. No-core fiber 2. Sureface plasmon resonance effect	5200 nm/RIU	7.2 nm/°C
[29] 2022	Cascaded two long period fiber gratings	−177.6 nm/RIU	0.1175 nm/°C
[30] 2022	1. D-shaped optic fiber with PDMS film 2. Tilted fiber Bragg grating	521.92 nm/RIU	4.38 nm/°C
This work	One silicon sore based fiber Bragg grating	208.76 dB/RIU	80.5 pm/°C

## Data Availability

The data utilized in this review article are available in the sources cited and can be accessed through web searches or by contacting the authors of the original studies.

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
