# Peer review of "Investigation of Silicon Core-Based Fiber Bragg Grating for Simultaneous Detection of Temperature and Refractive Index"

_sensors, 2023, doi:10.3390/s23083936_

Round 1

Reviewer 1 Report

The article titled "Investigation of Silicone core based FBG for simultaneously detected two parameters" by Yi-Lin Yu et al. is about a novel FBG which different characteristics then the typical one, with which is possible to sense temperature and refractive index simultanoeusly. The presented manuscript has merits but I have some comments:

- why in the title "Silicon" is with capital S?

- please improve the abstract (also from a grammatical point of view).

- in the introduction it is said that FBG is a low cost device, this sentences means nothing since the concept of "low cost" has to be compared with another similiar device. It doesn't seem to me that an electronic sensor is more expensive than an FBG sensor. Actually, the price of FBG sensors and the system to read them exacerbate their usage.

- again in the introduction it is said that due to the hughe temperature range (-100 to 700°C), this sensor can be used in "most of environment, e.g. nuclear reactor". Which is its behaviour if subjected to ionizing radiation?

- which is the typical length of the proposed FBG compared with the classical one?

- how about the strain sensitivity, is it enhanced or attenuated?

- how can you define a temperature and refractive index sensitivity just numerically? Do you expect the same values with experimental tests?

Author Response

Thanks for reviewer's question.

The attachment is our response to each question.

Reviewer 2 Report

Journal Name: Sensors-MDPI

Title: Investigation of Silicon core based FBG for simultaneously detected two parameters

In the current article K. Oguchi et. al., have developed the Silicon core-based FBG for sensing temperature and refractive index. The work is unique and suits MDPI – Sensors. The article is clear and the image quality is very good with proper explanation.

I recommend for a minor revision to publish at MDPI Polymers.  

Major revision

1.      I think authors can not use an abbreviation in the title to replace FBG from fiber Bragg grating

2.      In the title Instead of two parameters authors can mention temperature and refractive index

3.      At the introduction discuss what is the significance of simultaneous sensing of temperature and refractive index

4.      Maintain the same color for all axis labels (Fig. 6b)

5.      Citations are missing for equations

6.      At references kindly check the page numbers of each reference

7.      I think the authors can improve citations by citing recent articles (2022)

In short, I recommend the minor revision  

Author Response

(The authors gave the same response as above.)

Reviewer 3 Report

The authors present simulation work on a FBG scribed within a Silicon-core fiber. The simulation work focused in a potential application of such device as simultaneous temperature/RI sensing. Temperature sensing was allowed by means of RI changes in the FBG while RI sensing was allowed by means of RI changes at the tip of the device.

English language is very difficult to read. Some parts are cryptic and in general the line of though is unclear over the whole manuscript.

In the opinion of this reviewer this approach is an example of simulation over-use. Although different geometries and materials can be used to fabricate waveguides, the translation to real-life devices should always be emphasize. Moreover, analytical tools can give a more precise and elegant results on such a proposed device. In particular, solving the Helmholtz equation and then using convencional Bragg theory should give analytical expressions to play with and evaluate the role of core diameter, gratting period, etc.

Author Response

(The authors gave the same response as above.)

Reviewer 4 Report

Reviewer Comment

In this work, the authors have presented the theoretical study to sense the RI and temperature by using silicon core fiber-based FBG. The proposed work is suitable for the journal, but needs to be revised with the following comments:

1.      Abstract should need to be more elaborated to enhance the understanding of the work at first instance.

2.      “the core diameter of SCF needed to reduce as smaller than 1 μm when single mode operation,” could author explain it?

3.      Please explain why sensing capability of sensor is reduced while measuring both the parameters such as temperature and RI.

4.      We set the glass-based cladding diameter to 12 μm for decreasing the simulation period. What is the impact of this on propagation?

5.      Font size should be the same throughout the manuscript.

6.      As per the authors, the proposed structure is capable of sensing the RI. Is it possible to make this sensor useful in the field of biomolecules sensing.

7.      Authors are encouraged to discuss the practical implication of the proposed sensor.

8.      Authors are encouraged to cite some latest works which shows the RI based detection in different fields. Such as doi.org/10.1016/j.ijleo.2023.170516, doi.org/10.1016/j.hybadv.2022.100005

9.      If possible, please try to include a comparative study with some recent existing works.

Author Response

(The authors gave the same response as above.)

Reviewer 5 Report

The paper titled “Investigation of Silicon core based FBG for simultaneously detected two parameters” presents computational results on temperature, refractive index and simultaneous temperature and refractive index measurements using a silicon-based optical fiber.

First thing that caught my attention are the numerous cases of language misuse, such as wrong usage of single/plural form of the words, omission of verbs and prepositions, etc. I observed those throughout the manuscript, and although in some sentences the grammatical errors do not destroy the meaning, some parts of the manuscript is incomprehensible.

It is a common practice for the authors to reach a translation services that will revise scientific work and help to improve its quality.

Another significant comment is related to presenting a paper: introduction should contain problem statement; present the current research and the novelty of your work. What I see in the introduction are the random sentences from previous works that are combined together. Next, methodology section must provide the working principle of the sensor, what parameters are set, etc.  The paper titled “Investigation of Silicon core based FBG for simultaneously detected two parameters” did not provide its novelty, problem statement should be improved as well. 

Below is the list of comments that I would like to be addressed to the authors:

Line 13 “In this article, we theoretically and simulated the parameters of Silicon core fiber for single mode operation first.” – incorrect language usage, a verb is missing.

Line 17 “The proposed fiber sensor head can provide a method with simple and high sensitivity for various sensing target” – simple what? A noun is missing

Line 27 “The most frequently used system at the present time is the fiber Bragg grating (FBG) “- this statement, in my opinion, is controversial. FBGs have indeed been applied to various sensing methods, however does not make it the most frequently used system.  

Line 30 “Semiconductor based core fiber was becoming more and more popular “– please elaborate the term “popular”, this is not the scientific term.

Line 31 So far, the most researches of Silicon core based fiber was nonlinear effect – this sentence is not clear, what do you mean by saying that “most researches was nonlinear effect”?

Line 39 “In the reality, it needed to consider that how to connect SCF with fused silica-based fiber devices.” – English usage is not correct

Line 47 “Only considered the temperature sensing, the proposed fiber sensor head could be applied for temperature range from -100 to 700 °C, which means the sensor head could detect most of environment” – the whole sentence should be revised and rewritten

Line 73 “the length of Si-FBG was set 1200 μm”, please explain why such short size (1.2 mm) was chosen? And is it possible to perform any measurements with such a length of a fiber?

Line 85 In Equation (1) please describe the parameter “n”

Line 95 “which was because the smaller size” – the language usage is incorrect, “because of” should be written instead

Line 105 “there were still existed TE01 and TM01 mode simultaneously” -  the sentence is not correct in terms of language usage, please double check

Line 106 “when the operating wavelength shorter than cut-off wavelength” – the verb is missing, “the operating wavelength IS shorter than cut-off wavelength

Line 116 “which had a good spectral for sensing application” – both words “good” and “spectral” are adjectives, so the sentence is not clear what do you mean?

Line 116 “However, the smaller rc would cause insensitivity for the refractive index sensing, because the smaller  contact area between silicon core and environmental medium’’. This statement needs clarification: first of all, the core is not in contact with the environment, second, the sensing mechanism is based on the changes in the grating which will cause shifts in the reflected wavelength. So my question is how does the radius of the core affect the sensing performance of the fiber?

Line 137 “The reason why the trends not a linear but a squared relation was the material from of Silicon and fused silica were also have a squared related to wavelength.” This sentence is not written correctly. Moreover, the meaning of this statement is not clear at all and does not make any sense.  

Line 161 “In the temperature sensing, we set the boundary condition of fiber ends was the same of Silicon core fiber for ignoring the environmental refractive index condition.”- what kind of boundary conditions were applied in this work?

Line 175 In Fig. 5 (a) the paper presents the simulation output spectrum when applying different temperatures. What exact values were chosen for the calculations? Did you perform experiments as well, if yes what were the setup ?

Line 183 “The changes made a Fresnel reflection which likes a mirror, and Fresnel equation can be defined as”- may you please expand the idea, what exactly acts as a mirror, please add some details on the physics of this phenomenon

Line 250, Reference #2 He, R.; Teng, C.; Kumar, S.; Min, R. Review of high temperature measurement technology based on sapphire optical fiber. IEEE Sens. J. 2022, vol. 22, pp. 1081–1091. Here the title of the paper is not correct, in fact these authors published a paper titled “Polymer optical fiber liquid level sensor: A review”

Author Response

Thanks for reviewer's question.

We would to apologize our poor English grammar, and let you confuse and misunderstanding.

The attachment is our response to each question.

Round 2

Reviewer 3 Report

Although the manuscript has been substantially improved, this reviewer believes that several revisions must be covered before publication.

Minor comments:

Language usage is still poor. Just to mention a few examples:

Line 28: “FBG is one of powerful fiber devices…” I guess authors mean something like “FBG is a powerful fiber device”. However, this is only speculative.

Line 58: “…to out original research [19]. this condition…” Capital letter is missing.

Line 61-62: “However, the range for temperature and refractive index were still cover many sensing targets.” This idea is cryptic and must be revised.

Line 77: “…which was enough long for…” Authors may want to switch to “long enough”.

Line 189: “…the changes made a Fresnel reflection which likes a partially reflection mirror…” Must be revised.

-In Figure 2 the spatial profiles are truncated by black squares and triangles. It is not clear if the reason behind that is just to highlight each mode label or a truncation of the simulation. I recommend to show Figures as simple and informative as possible. This truncation is not helping to read easier the information within the Figure.

-Figure 4. Similar to Figure 2, Figure 4 lacks information about a potential truncation in the simulation. Moreover, the false-color scale is almost unintelligible. In addition, the text in lines 159-160 suggest that Fig. 4c depicts low reflection effects, while the caption in line 163 may suggest that Fig. 4c depicts high reflection effects.  

-All figures should be centered.

Major Comments:

1.       1. In lines 85-87 authors state:

“To investigate the condition, we simulated transmission transparency with different radius of silicon core from 0.45 to 0.75 and transmission mode, respectively, without fiber Bragg grating design.”

If I understand correctly, the condition is nothing more than single mode operation (line 84). My main concern of this manuscript is the necessity of finite-element simulation. Single mode condition can be readily studied by solving Helmholtz equation in cylindrical coordinates [R1].  More specifically, given the geometry and fiber material properties one can solve the so-called characteristic equation and obtain propagation constants for each mode.  The spatial profile can be also constructed based on Bessel functions and propagation constants.  Authors must state clearly why simulations tools are needed and which conditions are not covered by analytical resolution of Helmholtz equation.

2.       2. In lines 126-127 authors state:

“According to our records, the relationship between period and reflected central wavelength could be written as…”

Authors should elaborate on this. With “our records” authors mean literature? Or maybe previous experimental results? In any case, this point should be clarified and compared with reported literature.

3.       3. In refractive index measurements, a broader comparison of the proposed sensor with reported literature is needed. For example, all-fiber RI sensors based on Fourier transform have been reported previously [R2]. What are the advantages and disadvantages of the proposed sensor with previous reports?

4.       4. In the last phrase of the manuscript authors declare:

“Also, the analyzed method could be imported into machine learning, and develop artificial intelligence”.

This phrase is way too vague. Authors must elaborate if they mean to refine their predictions based on AI algorithms or advance in the area of IA as “…develop artificial intelligence” implies, or merely implement previously-developed IA algorithms.

R1. Saleh, B. E., & Teich, M. C. (2019). Fundamentals of photonics. john Wiley & sons. Chapter 10.

R2. Cuando-Espitia, N., Fuentes-Fuentes, M. A., May-Arrioja, D. A., Hernández-Romano, I., Martínez-Manuel, R., & Torres-Cisneros, M. (2021). Dual-point refractive index measurements using coupled seven-core fibers. Journal of Lightwave Technology, 39(1), 310-319.

Author Response

To reviewer

Thank you for giving us the comments, and reply your question one by one as the attachment.

Reviewer 4 Report

not applicable

Author Response

I did not see any comment here.

However, thanks for reviewer's previous suggestions and questions.

Round 3

Author Response

Q1. Authors should clarify the use of the “monomodal operation” term along the manuscript.

Reply

Thanks for reviewer’s suggestion.

We revised our sentence as below

  1. Line 11-12 (Abstract)

We first discussed the parameters of Silicon core fiber for single mode operation.

“We first discussed the parameters of Silicon core fiber for near single mode operation.”

  1. Line 110-112

The red solid line is the limitation for single mode operation which means the V-number below the red solid line is necessary for single mode, and the value was around 6.4.

“The red solid line is the limitation for near single mode operation which means the V-number below the red solid line is necessary for near single mode, and the value was around 6.4.”

  1. Line 113-114

However, the TM01 mode was operated with higher loss than TE01 mode, specially in smaller rc cases, which was described in Figure 2(f).

However, TE01 mode was more efficient transmission than TM01 due to the lower loss, which was so called near single mode operation in this article. Figure 2(f) shows the transmission efficiency with different rc and modes.

After we discussion, we revised the description single mode operation to near single mode operation, because there were still TM01 mode generated. However, TE01 mode was more efficient transmission than TM01 due to the lower loss.

About why we use software to simulate the issue, because we did not want to insert too much mathematics and calculations in the manuscript.